# Ready for Action! Destination Climate Change Communication: An Archetypal Branding Approach

**DOI:** 10.3390/ijerph20053874

**Published:** 2023-02-22

**Authors:** Uglješa Stankov, Viachaslau Filimonau, Miroslav D. Vujičić, Biljana Basarin, Adam B. Carmer, Lazar Lazić, Brooke K. Hansen, Danijela Ćirić Lalić, Dino Mujkić

**Affiliations:** 1Department of Geography, Tourism and Hotel Management, Faculty of Sciences, University of Novi Sad, 21000 Novi Sad, Serbia; 2Centre for Sustainability and Wellbeing in the Visitor Economy, School of Hospitality and Tourism Management, University of Surrey, Guildford GU2 7XH, UK; 3School of Hospitality and Tourism Management, Muma College of Business, University of South Florida, Tampa, FL 33620, USA; 4Department of Industrial Engineering and Management, Faculty of Technical Sciences, University of Novi Sad, 21000 Novi Sad, Serbia; 5Faculty of Sports and Physical Education, University of Sarajevo, 71000 Sarajevo, Bosnia and Herzegovina

**Keywords:** climate change communication, climate action, destination branding, archetypal branding, sustainable tourism

## Abstract

At the destination level, destination branding may coexist with climate change communication. These two communication streams often overlap because they are both designed for large audiences. This poses a risk to the effectiveness of climate change communication and its ability to prompt a desired climate action. The viewpoint paper advocates the use of archetypal branding approach to ground and center climate change communication at a destination level while concurrently maintaining the uniqueness of destination branding. Three archetypes of destinations are distinguished: villains, victims, and heroes. Destinations should refrain from actions that would make them appear to be climate change villains. A balanced approach is further warranted when portraying destinations as victims. Lastly, destinations should aim at assuming the heroic archetypes by excelling in climate change mitigation. The basic mechanisms of the archetypal approach to destination branding are discussed alongside a framework that suggests areas for further practical investigation of climate change communication at a destination level.

## 1. Introduction

Both the scientific community and society are paying more attention to how to communicate climate change. Climate change communication has developed utilizing a variety of communication techniques and strategies alongside more established communication streams such as health, risk, and science communication [1]. The aim of climate change communication is to educate, inform, warn, persuade, and mobilize recipients [2]. Climate change communication is a complex process extending beyond simple concerns about the types of messages and messengers or communication channels. Climate change communication is largely shaped by the characteristics of the audiences, contexts, and goals of communication together with the mutual interactions and dynamics between parties in those communications [3]. In the context of tourism, climate change communication should ultimately result in climate action, seen as an effort to reduce greenhouse gas (GHG) emissions, while also strengthening the adaptive capacity of destinations for climate-related impacts [4]. Climate change communication is gaining prominence as a component of overall tourism marketing communication [5,6] in conjunction with destination branding [7]. Recent movements including Tourism Declares a Climate Emergency [8] and the COP26 Glasgow Declaration on Climate Action in Tourism [9] advocate taking a public stance on climate action and promoting climate action as part of tourism offerings. 

Destination branding occupies a crucial position in tourism marketing; as such, it is often defined as an aspect of place branding in which the place entity is viewed from the perspective of tourists and the tourism industry [10,11,12]. Tourism academics are typically interested in topics such as brand identity, image, and personality and the political dynamics of place branding, the embodiment of heritage into branding, communicating brands in media, the interaction between country-of-origin branding and destination branding, and the relationship between branding and regional development and regeneration [10]. Within the destination branding approaches, the concept of brand personality and its impact on consumer behavior has emerged as a focal research topic [13,14]. Usually, consumers embrace brands with strong, positive personalities. An example of the idea is the Jeep brand. Jeep embodies clear personality traits of seeking adventure, toughness, and a strong interest in the outdoors [15] due to the natural human tendency to anthropomorphize nonhuman objects [16]. Typically, brand personality is referred to as a destination [17] or a product [15] and builds on one set of characteristics best suited to the object of branding or the image that needs to be projected [18]. For example, the perceived destination personality of Las Vegas is multi-dimensional, including vibrancy, sophistication, competence, contemporary, and sincerity [19].

At the destination level, climate change communication occurs concurrently with destination branding [20], acting as an additional layer of communication [21]. In fact, in order to keep pace with global and local sustainability and green marketing initiatives and to contribute to the “green” transition, many destinations have acknowledged their obligation to become more environmentally sustainable [22]. For instance, the Eggental Valley in Italy promotes itself as the South Tyrol’s Sustainable Dolomites region, which is more than just a tourist destination but a region where corporate decisions are informed by their consequences for profit, the environment, and people. By collaborating closely with the Global Sustainable Tourism Council, they branded the region as a “green” destination, working directly with stakeholders, communities, and the local population in addition to launching a number of sustainable-related initiatives, including easy-to-use carbon emissions calculators for visitors [23]. From the practical perspective, both destination branding and climate change communication rely on activities such as advertising, public relations, or social media efforts that are used to provide information and communicate with customers [24]. A common ground for destination branding and climate change communication is that both are mainly orientated toward mass audiences. However, this poses challenges for both endeavors in terms of message clarity, complexity, efficiency, and penetration as well as in terms of directing messages to induce climate action.

While some destinations portray that they are immediately threatened by climate change (e.g., some Pacific islands), others experience climate change effects as accidental but more frequent occurrences [25] for example, in the case of floods of 2021 in Germany [26]. Some destinations even view climate change as an opportunity by promoting longer, hotter seasons or “last chance tourism” (e.g., polar bear viewing industry in the Arctic) [27]. Even though there have been attempts to integrate climate change communication into tourism marketing and destination branding efforts [7,24], the question of how to operationalize this integration effectively remains pressing [28]. Importantly, paradoxical and confusing destination branding (for example, consumers require energy-intensive modes of transportation to reach destinations marketed as environmentally friendly or as carbon-neutral destinations, such as in the case of ecotourism from Europe to Costa Rica) can hamper the capacity of climate change communication to inspire successful climate change actions and readiness [29]. In addition, when climate change communication is aligned with destination branding, the risk of brand schizophrenia (i.e., multiple brand personalities) exists [30]. Typically, brand schizophrenia is viewed as a threat to brand identity that may be a coherent aspect in attaining a competitive advantage [30,31]. In terms of destination branding, brand schizophrenia may occur when various sub-brands with distinct stakeholders are not integrated into a single regional brand [32], or there is no commonplace attachment for those stakeholders [33].

Due to the varied and sometimes paradoxical place that destinations have regarding climate change (i.e., some destinations are perceived as victims of climate change, some contribute significantly to climate change, while some invest in climate change mitigation and adaptation), communicating climate change messages is multifaceted. Due to these circumstances, current destination brands should be able to incorporate an additional layer of engaged climate change messages while maintaining their main perceived values and distinctive brand characteristics. The motivation for this viewpoint is to contribute to destination branding when needing to incorporate climate change communication. Authors believe that in a time of increased competition [34], destination brands looking for simplicity [35], and general information overload in the digital era [36], environmentally responsible destination branding should seek for more engaging ways to shape brand meaning and value, transcending traditional definitions of the brands by embodying cultural, ideological, and psychological values and providing representational and rhetorical power [37]. The viewpoint argues that destination marketing managers may consider adopting an archetypal approach [38] to ground and focus climate change communication and maintain or enhance existing destination branding. 

Archetype is defined as a constantly recurring symbol in literature, painting, or mythology, for example, a superhero [39]. With the help of an archetype serving as the center of all communications and fostering a stronger bond with customers, archetypal brands may build an improved resonant brand [40]. With an archetypal approach, the customer’s brand meaning, and experience are the primary considerations, while the physical aspect of a product or service provides a means to accomplish a desired brand meaning [38]. 

The text that follows is intended to stimulate scholarly debate and motivate the pragmatic investigation of the archetype branding approach when incorporating climate change communication in destination branding. 

## 2. Avoiding the Villain Archetype 

At the Energy and Man Symposium in 1959, celebrating the 100th year for the oil industry, Edward Teller, known as the father of the hydrogen bomb, decreed that oil needed to be supplemented by other energy sources. Teller made clear that the use of fossil fuels was contaminating the atmosphere and would lead to increased global warming by an estimated one to five degrees Fahrenheit [41]. Fossil fuel (oil) companies understood the global damage that would be caused by their products. For instance, a global oil and gas corporation predicted global warming in 1982 with an in-house climate model, but no meaningful actions to mitigate global warming were taken [42]. The key global CO_2_ emitters were subsequently portrayed as villains in the global climate change narrative. Eventually, big energy companies adopted a more optimistic tone and suggested innovative ways to combat climate change while maintaining profitable business operations in a carbon-constrained society [43]. Despite the efforts undertaken by the industry to present itself as a carbon-change mitigator, the public found it difficult to embrace this rebranding of the oil and gas industry. 

Although the complex interrelationships between tourism and human-caused climate change have been extensively studied [20,44,45,46], and despite the fact that climate change communication as an academically and practically important domain is growing in significance within the tourism domain [6,47], unresolved issues remain that can attribute the role of villain to the whole tourism industry. 

International tourism contributes substantially to climate change because it accounts for up to 8% of the human induced GHGs [48,49]. Transport-related carbon emissions are the main contributor. Most of the carbon impact is attributed to air travel to and from destinations [4]. A combination of short- to medium-distance local travel stays in luxury hotels and regular on-site energy-intensive activities of tourists may generate large GHG emissions at a destination, thus partially or fully offsetting the GHG emissions from transport to a destination [50]. 

Even though direct parallels cannot be drawn between the case of the oil and gas industry and destinations (the former generates GHG emissions directly, while destinations contribute to climate change indirectly), the risks associated with climate change communication are also present. As seen in the example of the oil and gas industry, where the division of roles is straightforward (i.e., the oil and gas industry as villains and the environment as victim), the tourism industry, however, represents a more complicated scenario. Tourism heavily contributes to the increased use of fossil fuels, which directly endangers the environment of destinations in which tourism operates [51]. This requires tourists to re-think if they want to pursue their hedonistic travel motivation and contribute to climate change with irresponsible holiday choices [52]. Concurrently, the more climate-benign alternatives and initiatives offered by the tourism industry still operate outside of the realm of mass tourism [53]. 

Tourism marketing actions tend to adopt ambitious and complex roles of pro-sustainability communication, including climate change communication, while attempting to avoid, whenever possible, a conflict with their own agendas and raison d’être, which, in many instances, are not genuinely concerned with the issue of climate change [7]. In addition, destination marketing is not immune to adopting various environmental public discourses even as a gimmick for short-term profits [54]. In the case of climate change communication, however, the stakes are much higher compared to the relatively limited effects of some greenwashing routines on reduced consumer satisfaction that can occur in the tourism industry [55,56]. By superficially adopting climate change communication, destinations can easily fall into a trap of becoming an untrustworthy source of information. Destinations may undermine their source credibility [57], thus threatening the achievement of goals of climate change communication.

## 3. A Victimized Position 

In projecting a victim archetype, destinations pinpoint the situation in which they are victimized by climate change. For tourists, it can be disturbingly enlightening to realize that their favorite destination may vanish. Sinking island states in the Pacific [58], disappearing Alpine glaciers [59], wildfires in the Mediterranean [60], coral reef bleaching in Australia [61], and the dramatic floods of July 2021 in Germany and neighboring countries [26] are some of the widely known devastating effects of climate change on destinations. 

These consequences of climate change can prompt protective behavior among certain segments of consumers [62], including a wish to change their behavior [63] and support for tourists’ ethical consumption [64] in order to help destinations or at least to understand and appreciate the measures undertaken by destinations to mitigate or adapt to climate change. If the direct or indirect actions in helping endangered destinations are not altruistic acts to provide support to a perceived victim but are motivated by individual selfishness to visit, it is still considered as socially desirable (i.e., if we do not go, they will not benefit from our money) [65]. For example, Venice, together with other iconic tourism cities such as New Orleans or Miami, located in coastal areas, are likely to suffer from climatic changes more than other destinations [66]. In the case of Venice and its plans for additional tourist taxation for day-trippers to mitigate over-tourism [67], portraying vulnerabilities of the city and presenting it as a victim can enhance an understanding of the problem among tourists. This message can be coupled with other risks related to climate change, among which the rising sea level is the most severe [66]. 

Here, it must be emphasized that the projection of the victimized position should be precisely balanced for a destination. The destination branding, which often considers establishing a positive image, may suffer if the victim archetype is assigned in its entirety. The destination should indicate the extent to which presenting a victimized position is reasonable for customers to avoid coming across as a helpless and hopeless case. 

## 4. A Hero Archetype 

Many tourists are not persuaded by the relationship between human-caused climate change and tourism and the urgency of taking climate action because of personal prejudices, their world views, or a focus on profit [68]. Skepticism, denial, lack of interest or awareness [65,69], and the attitude–behavior gap in sustainable consumption [70] are present in all parts of the tourism industry, from mass tourism consumers to smaller parties that make up a complex web of tourism providers at a destination [71].

According to the UNWTO, tourism must establish its own “high-ambition scenario”, one that envisions a transition to low-emission and highly effective operations [4]. To this end, many destinations present themselves as leaders in climate change mitigation, adaptation, and innovation [72] and strive to become net-zero destinations [73]. Some of these destinations, partly due to tourism development, still have high carbon footprints, but they actively work to change and find more sustainable solutions [51]. In essence, by promoting successful stories and endeavors in green transition, the destinations broadcast messages that evoke hope (i.e., hope appeals) [74]. This is aligned with the views that destination management must lead by example in terms of educating tourists about climate change and implementing mitigation measures [24]. For instance, the city of Las Vegas and Clark County, in which it is located, release large GHGs because of energy consumption necessary to run factories, power buildings, and run transportation. Knowing the influence of GHGs on climate change and being faced with increasing threats that extreme weather events bring to Las Vegas, massive investments are made to promote green solutions (for example, building bike lanes, switching streetlights to LED lighting, and powering public buildings, parks, and traffic lights with renewable energy) [75]. Understanding Clark County’s efforts are not well recognized and acknowledging that discussing climate change can be unpleasant in a leisure context, Las Vegas has created a guide for talking about climate change with family and friends over holidays and downscaling it to individual levels [76]. This and other similar examples portray a high potential in promoting existing climate actions and destination branding under the hero archetype branding to amplify and provoke more climate actions among different tourism stakeholders. By projecting more destinations as heroic archetypes on a global scale, it may be feasible to paint a favorable picture of the entire tourism industry as one that supports the environment [77] and overall sustainable development goals promoted by the United Nations [72] as well as an industry that embraces innovation [78] and alternative modes of development [79]. 

## 5. A Framework for Empirical Research on Climate Change Communication at a Destination Level Based on Archetypal Branding 

By applying archetypal branding to a destination, in which a brand is not only defined as a set of ideals (such as youthful or elegant) but also as an archetypal personality, a more emotional appeal could be generated around them [80]. Since the media are the main channel from which most people obtain information, the media play an important role in how people think about climate change [29,81]. Attaching the right story to a destination brand that builds on well-known stereotypes (i.e., hero or victim) from the public culture and mass media [80] can cause different reactions from tourists based on a complex mix of emotional and cognitive triggers. 

In order to stimulate the desired climate actions among broad audiences, a complex set of mutual relationships between destinations and climate change should be understood to define the primary set of archetypal roles that destinations may play (i.e., victim or hero). Assigning a clear archetypal role should act as a stimulus evoking a response. Indeed, a destination-level stimulus coupled with the climatic characteristics of climate-dependent tourism markets is becoming important in tourists’ decision-making process [48,65,82], particularly for the environmentally responsible behavior of tourists [57]. 

While most strategies for communicating climate change rely on cognitive motivation to encourage pro-environmental behavior, an ample amount of research focuses on the role of emotions in causing pro-environmental advocacy behavior [62,83], arguing for its higher effectiveness over strictly factual information presentations [84], for example, as given by climate change models. In the case of archetypal branding, emotions are the byproducts of typical archetypal personalities [85] associated with destination brands/personalities. However, emotions are not viewed as simple levers to be pulled to achieve desired results but rather as an essential part of a cognitive feedback system [86] connected to a particular destination brand that should lead to more meaningful and desirably focused responses. Here, the purpose of the inclusion of archetypal branding in this process is to provoke more common inner reactions and resultant behavior to external environmental stimuli based on the stimulus-organism-response theory. For instance, the desired responses from the audience can be to endorse (a hero), to help/support (a victim), and to avoid/punish (a villain) (Figure 1). 

Even in the same cultural settings, the perceptions of brand personality [16] and/or archetypes are hardly universal [87]. The audience for destinations and their climate change messages are wide, ranging from tourists as common recipients to numerous players on the tourism offer side (see Figure 1). In the case of tourists, how they absorb climate change messaging will be determined not only by their personal traits but also by the setting of their travel (e.g., island, urban destination, or rural destination), which includes but is not limited to travel type (e.g., business or leisure), motivation (e.g., hedonic or eudemonic), and the level of perceived climate emergency. 

Apart from the typical goals of climate change communication (informing, educating, warning, persuading, and mobilizing audiences), the ultimate goal is to ensure meaningful climate change actions among the majority of visitors to a certain destination as well as among other tourism stakeholders (e.g., destination management/marketing organization, travel supply chain, and formal and informal tourism education). The purpose of the archetypal branding approach in this case is also to allow for the peaceful coexistence of diverse roles that the same destination can play in terms of climate change, hence keeping the destination’s brands coherent. As such, the archetypal branding approach should express the complexities of climate change communication in a meaningful way to general audiences, contributing to communication goals while respecting and adapting to the communication context. 

## 6. Conclusions

This viewpoint proposes archetypal branding approach as a means of incorporating climate change communication in destination branding. This approach may potentially inspire more varied climate change actions while being aligned with destination demands, scientific support, and intended destination branding. In particular, an archetypal branding approach can encourage scholars to undertake studies on segmenting destinations in line with the extent of their engagement/disengagement in climate change mitigation. An archetypal branding approach can also prompt scholars to more actively study destinations from the perspective of their pro-sustainability branding. For example, scholars can help destination managers to frame messages on how they (do not) tackle the major societal challenge of climate change, thus aiding destinations and their decision makers in promoting themselves to tourist groups with various degrees of climate change sensitivity. Ultimately, an archetypal branding approach can shape an agenda for future scholarly discourse on the role of branding for destinations in a time of rapidly changing climate, thus contributing to the ongoing scientific debate on the complex relationships between tourism and the environment.

It is difficult to translate the scientific understanding of climate change into meaningful actions and policies in general; this is true across many industries, not just tourism. This discrepancy between scientific understanding and actual practice is commonly attributed to the mass media’s ineffective public communication regarding climate change [6]. Here, destination branding may be understood as a means of connecting the science of climate change to a large number of tourists with an instantly recognizable message. 

To assist localized initiatives and approaches in communicating climate change, destination branding needs more efficient and adaptable methods. While the information on carbon footprint of certain travel and transportation services becomes more widely available (e.g., when choosing flights), together with the provided ability to act (i.e., to pay an environmental tax when visiting a destination), climate change communication incorporated into destination branding still needs novel approaches to boost its awareness. Archetypal branding can play an important role. For instance, archetypal branding is aligned with the views that the most effective form of communicating climate science to large audiences is non-persuasive communication [88] of factual information that is frequently repeated by reliable sources (in this case, the destinations presented and perceived as the hero archetypes may seem more trustworthy to tourists), bridging the perception gap as well as the divide between the audience types [89]. 

In order to move away from purely theoretical assumptions, an empirical examination of archetypal branding as part of overall destination branding is needed. It is uncommon to relate archetypal branding to places, as it has traditionally been associated with products and services rather than destinations. Developing archetypal branding requires a mix of a complex set of data, ranging from the geographical characteristics of a destination to its socioeconomic situations, tourist profiles [90], and climate change requirements and initiatives [36,91]. This significantly broadens the scope of empirical study in destination branding both in terms of developing non-traditional branding strategies such as the archetypal approach and adding another layer of climate change communication. 

## Figures and Tables

**Figure 1 ijerph-20-03874-f001:**
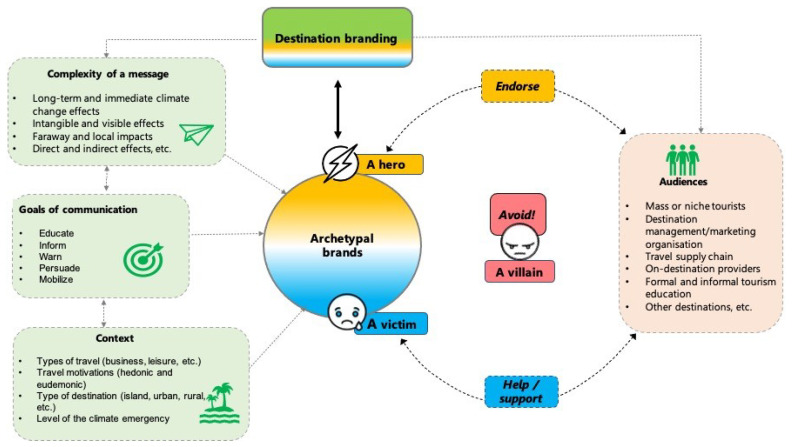
A framework for climate change communication at a destination level based on archetypal branding.

## Data Availability

Not applicable.

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
