# Peer review of "Ready for Action! Destination Climate Change Communication: An Archetypal Branding Approach"

_ijerph, 2023, doi:10.3390/ijerph20053874_

Round 1
Reviewer 1 Report
The main idea of this manuscript is interesting and authentic, but it was not declared by the authors as a “Discussion” or “Short Communication”. The authors state that the intention of their manuscript is to “stimulate scholarly debate and motivate pragmatic use of the archetype branding strategy in a destination's climate change communication”. The main weakness of the manuscript is therefore related to the complete lack of a research approach. In consequence, the manuscript cannot be considered and analyzed as an “Article” because it only introduces a potential strategic idea related to using the three brand archetypes in a destination's climate change communication. In the presentation of article types that can be submitted to IJERPH, it is stated that “Article”-type manuscripts should be based on “original research”, with results that provide substantial amount of new information. It is not the case of the present manuscript, because, as I wrote before, there is no research in it!
Author Response
RESPONSE TO REVIEWERS
Dear colleagues,
Thank you for your feedback. We have considered all the comments highlighted by the reviewers and revised the manuscript accordingly. Please find our responses to the reviewers’ comments below.
Reviewer 1
Comment: The main idea of this manuscript is interesting and authentic, but it was not declared by the authors as a “Discussion” or “Short Communication”. The authors state that the intention of their manuscript is to “stimulate scholarly debate and motivate pragmatic use of the archetype branding strategy in a destination's climate change communication”. The main weakness of the manuscript is therefore related to the complete lack of a research approach. In consequence, the manuscript cannot be considered and analyzed as an “Article” because it only introduces a potential strategic idea related to using the three brand archetypes in a destination's climate change communication. In the presentation of article types that can be submitted to IJERPH, it is stated that “Article”-type manuscripts should be based on “original research”, with results that provide substantial amount of new information. It is not the case of the present manuscript, because, as I wrote before, there is no research in it!
Answer: We thank the reviewer for recognizing the authenticity of our idea. We fully understand and accept the critics of our paper being conceptual in nature. Our intention was to write a viewpoint paper and highlight the main elements for further conceptualisation of destination climate change communication based on archetypal branding.
IJERPH does not explicitly specify if they want to publish conceptual papers. In their guide for authors, they state that they publish ‘all original research manuscripts provided that the work reports scientifically sound experiments and provides a substantial amount of new information’. We hope you will agree that not all studies are based on experiments. We also hope you will agree that our paper does provide new information as related to destination climate change communication based on archetypal branding.
In the revised version it is now clearly stated that this is a viewpoint paper. Now we better articulate the motivation of the paper, as follows: “... is intended to stimulate scholarly debate and motivate the pragmatic investigation of the archetype branding approach when incorporating climate change communication in destination branding”.
The reviewer also pinpointed in the checklist that cited references relevant to the research could be improved. We added the following:
Bassols, N., & Leicht, T. (2020). Exploring destination brand disengagement in a top-down policy context: Lessons learned from Cartagena, Colombia. Journal of Place Management and Development, 13(3), 347–363. https://doi.org/10.1108/JPMD-06-2019-0040
Blain, C., Levy, S. E., & Ritchie, J. R. B. (2005). Destination Branding: Insights and Practices from Destination Management Organizations. Journal of Travel Research, 43(4), 328–338. https://doi.org/10.1177/0047287505274646
Future Strategy Eggental 2030. (2022). Eggental 2030: The future strategy of the Dolomites holiday region. TOGETHER WE ARE... Future Strategy Eggental 2030 A Participative Strategy. https://eggental.com/en/eggental/strategy-eggental-2030
Hankinson, G. (2005). Destination brand images: A business tourism perspective. Journal of Services Marketing, 19(1), 24–32. https://doi.org/10.1108/08876040510579361
Hanna, S., Rowley, J., & Keegan, B. (2021). Place and Destination Branding: A Review and Conceptual Mapping of the Domain. European Management Review, 18(2), 105–117. https://doi.org/10.1111/emre.12433
Kotsi, F., Balakrishnan, M. S., Michael, I., & Ramsøy, T. Z. (2018). Place branding: Aligning multiple stakeholder perception of visual and auditory communication elements. Journal of Destination Marketing & Management, 7, 112–130. https://doi.org/10.1016/j.jdmm.2016.08.006
Pongsakornrungsilp, P., Pongsakornrungsilp, S., Jansom, A., & Chinchanachokchai, S. (2022). Rethinking Sustainable Tourism Management: Learning from the COVID-19 Pandemic to Co-Create Future of Krabi Tourism, Thailand. Sustainability, 14(18), Article 18. https://doi.org/10.3390/su141811375
Pongsakornrungsilp, S., Pongsakornrungsilp, P., Pusaksrikit, T., Wichasin, P., & Kumar, V. (2021). Co-Creating a Sustainable Regional Brand from Multiple Sub-Brands: The Andaman Tourism Cluster of Thailand. Sustainability, 13(16), Article 16. https://doi.org/10.3390/su13169409
Pongsakornrungsilp, S., & Schroeder, J. E. (2017). :How consumers co-create. In M. Keller, B. Halkier, & T. A. Wilska (Eds.), Routledge Handbook on Consumption. Routledge.
Stankov, U., & Gretzel, U. (2021). Digital well-being in the tourism domain: Mapping new roles and responsibilities. Information Technology & Tourism, 23(1), 5–17. https://doi.org/10.1007/s40558-021-00197-3
Xara-Brasil, D., Miadaira Hamza, K., & Marquina, P. (2018). The meaning of a brand? An archetypal approach. Revista de Gestão, 25(2), 142–159. https://doi.org/10.1108/REGE-02-2018-0029
Zenker, S., Braun, E., & Petersen, S. (2017). Branding the destination versus the place: The effects of brand complexity and identification for residents and visitors. Tourism Management, 58, 15–27. https://doi.org/10.1016/j.tourman.2016.10.008
Reviewer 2 Report
This is an interesting conceptual paper by focusing on destination branding and climate change communication. However, I believe that authors can improve the quality of the paper through these suggestions:
1. Please state the problems or issues related to destination branding and climate change communication why the authors would like to propose a new approach. Authors might need to decide whether this paper is about destination branding or climate change communication. Each of them has different points and also components.
2. Authors need to consider the difference between branding and communication. The first is branding and destination brand, and the latter is communication and climate change issues. What do you want to develop the concept? It is interesting if the authors develop the destination brand in terms of climate change or sustainability.
3. However, if you are developing destination brand with the climate change communication, I would recommend authors to see these papers:
Bassols, N.; Leicht, T. Exploring destination brand disengagement in a top-down policy context: Lessons learned from Cartagena, Colombia. J. Place Manag. Dev. 2020, 13, 347–363.
Kotsi, F.; Balakrishnan, M.S.; Michael, I.; Ramsøy, T.Z. Place branding: Aligning multiple stakeholder perception of visual and auditory communication elements. J. Dest. Mark. Manag. 2018, 7, 112–130.
Pongsakornrungsilp, S.; Schroeder, J.E. Consumers and Brands: How Consumers Co-Create Value. In Routledge Handbooks of Consumption; Keller, M., Halkier, B., Wilska, T.A., Eds.; Routledge: New York, NY, USA, 2017; pp. 89–101.
Pongsakornrungsilp, S; Pongsakornrungsilp, P.; Pusaksrikit, T; Wichasin, P.; Kumar, V. Co-Creating a Sustainable Regional Brand from Multiple Sub-Brands: The Andaman Tourism Cluster of Thailand. Sustainability, 2021, 13(16), 9409. https://doi.org/10.3390/su13169409.
Zenker, S.; Braun, E.; Peterson, S. Branding the destination versus the place: The effects of brand complexity and identification for residents and visitors. Tour. Manag. 2017, 58, 15–27.
4. Why do we need an archetypal branding approach? What are the problems to put scholars thinking about a new approach? Authors might need to discuss the branding concepts and communication models in order to point out the gap or issue for requiring a new approach. Authors need to see different branding concepts, especially destination branding which there are many components and participants in the branding process. In destination branding, there are many stakeholders related to the branding process, and authors need to realize the difference between the branding process of destination branding and product branding.
5. Authors need to clarify why an archetypal branding approach can help scholars to understand destination branding process or can build destination brand in terms of villain, victim and hero.
Author Response
RESPONSE TO REVIEWERS
Dear colleagues,
Thank you for your feedback. We have considered all the comments highlighted by the reviewers and revised the manuscript accordingly. Please find our responses to the reviewers’ comments below.
Reviewer 2
Comment: This is an interesting conceptual paper by focusing on destination branding and climate change communication. However, I believe that authors can improve the quality of the paper through these suggestions:
- Please state the problems or issues related to destination branding and climate change communication why the authors would like to propose a new approach. Authors might need to decide whether this paper is about destination branding or climate change communication. Each of them has different points and also components.
Answer: We thank the reviewer for the positive outlook and suggestions. Our article focuses on destination branding that could also be used to communicate climate change information as part of global or local sustainable marketing initiatives. This is now better clarified in the Introduction.
“The motivation of this viewpoint is to contribute to destination branding when needing to incorporate climate change communication. Authors believe that in the time of increased competition (Bassols & Leicht, 2020), destination brands looking for simplicity (Zenker et al., 2017), and general information overload in the digital era (Stankov & Gretzel, 2021), environmentally-responsible destination branding should seek for more engaging ways to shape brand meaning and value, transcending traditional definitions of the brands, by embodying cultural, ideological, and psychological values, and providing representational and rhetorical power (Pongsakornrungsilp & Schroeder, 2017). The viewpoint argues that destination marketing managers may consider adopting an archetypal approach (Xara-Brasil et al., 2018) to ground and focus climate change communication and maintain or enhance existing destination branding.“
Comment: 2. Authors need to consider the difference between branding and communication. The first is branding and destination brand, and the latter is communication and climate change issues. What do you want to develop the concept? It is interesting if the authors develop the destination brand in terms of climate change or sustainability.
Answer: The text has been rewritten to pinpoint the focus on destination branding and climate change communication. We have rewritten the goal of the paper to better reflect upon our goal of assisting destination branding to incorporate climate change communication and build archetypal destination brands.
Comment: 3. However, if you are developing destination brand with the climate change communication, I would recommend authors to see these papers:
Bassols, N.; Leicht, T. Exploring destination brand disengagement in a top-down policy context: Lessons learned from Cartagena, Colombia. J. Place Manag. Dev. 2020, 13, 347–363.
Kotsi, F.; Balakrishnan, M.S.; Michael, I.; Ramsøy, T.Z. Place branding: Aligning multiple stakeholder perception of visual and auditory communication elements. J. Dest. Mark. Manag. 2018, 7, 112–130.
Pongsakornrungsilp, S.; Schroeder, J.E. Consumers and Brands: How Consumers Co-Create Value. In Routledge Handbooks of Consumption; Keller, M., Halkier, B., Wilska, T.A., Eds.; Routledge: New York, NY, USA, 2017; pp. 89–101.
Pongsakornrungsilp, S; Pongsakornrungsilp, P.; Pusaksrikit, T; Wichasin, P.; Kumar, V. Co-Creating a Sustainable Regional Brand from Multiple Sub-Brands: The Andaman Tourism Cluster of Thailand. Sustainability, 2021, 13(16), 9409. https://doi.org/10.3390/su13169409.
Zenker, S.; Braun, E.; Peterson, S. Branding the destination versus the place: The effects of brand complexity and identification for residents and visitors. Tour. Manag. 2017, 58, 15–27.
- Why do we need an archetypal branding approach? What are the problems to put scholars thinking about a new approach? Authors might need to discuss the branding concepts and communication models in order to point out the gap or issue for requiring a new approach. Authors need to see different branding concepts, especially destination branding which there are many components and participants in the branding process. In destination branding, there are many stakeholders related to the branding process, and authors need to realize the difference between the branding process of destination branding and product branding.
Answer: We highly appreciated the suggested literature and we have used it to strengthen the theory on destination branding through the whole Introduction section and to highlight why we need an archetypal branding approach. The following parts are added:
“Destination branding occupies a crucial position in tourism marketing, as such it is often defined as an aspect of place branding in which the place entity is viewed from the perspective of tourists and the tourism industry (Blain et al., 2005; Hankinson, 2005; Hanna et al., 2021). Tourism academics are typically interested in topics such as brand identity, image, and personality, the political dynamics of place branding, the embodiment of heritage into branding, communicating brands in media, the interaction between country-of-origin branding and destination branding, and the relationship between branding and regional development and regeneration (Hanna et al., 2021).”
“In fact, in order to keep pace with global and local sustainability and green marketing initiatives and to contribute to the ‘green’ transition, many destinations have acknowledged their obligation to become more environmentally sustainable (Pongsakornrungsilp et al., 2022). For instance, the Eggental valley in Italy promotes itself as the South Tyrol's Sustainable Dolomites region, which is more than just a tourist destination, but a region where corporate decisions are informed by their consequences for profit, the environment, and people. By collaborating closely with the Global Sustainable Tourism Council, they branded the region as a ‘green’ destination, working directly with stakeholders, communities, and the local population, and launching a number of sustainable-related initiatives, including easy-to-use carbon emissions calculators for visitors (Future Strategy Eggental 2030, 2022).”
“In addition, when climate change communication is aligned with destination branding, the risk of brand schizophrenia (i.e., multiple brand personalities) exists [29]. Typically, brand schizophrenia is viewed as a threat to brand identity that may be a coherent aspect in attaining a competitive advantage [29,30]. In terms of destination branding, brand schizophrenia may occur when various sub-brands with distinct stakeholders are not integrated into a single regional brand (Pongsakornrungsilp et al., 2021) or there is no commonplace attachment for those stakeholders (Kotsi et al., 2018).”
Comment: 5. Authors need to clarify why an archetypal branding approach can help scholars to understand destination branding process or can build destination brand in terms of villain, victim and hero.
Answer: Thank you for this comment. The following passage has been added to the revised section 4. Concluding remarks:
This viewpoint proposes an archetypal branding approach as a means of incorporating climate change communication in destination branding. This approach may potentially inspire more varied climate change actions while being aligned with destination demands, scientific support, and intended destination branding. In particular, an archetypal branding approach can encourage scholars to undertake studies on segmenting destinations in line with the extent of their (dis)engagement in climate change mitigation. An archetypal branding approach can also prompt scholars to more actively study destinations from the perspective of their pro-sustainability branding. For example, scholars can help destination managers to frame messages on how they (do not) tackle the major societal challenge of climate change, thus aiding destinations and their decision-makers in promoting themselves to tourist groups with various degrees of climate change sensitivity. Ultimately, an archetypal branding approach can shape an agenda for future scholarly discourse on the role of branding for destinations in a time of rapidly changing climate, thus contributing to the ongoing scientific debate on the complex relationships between tourism and the environment.
References:
Bassols, N., & Leicht, T. (2020). Exploring destination brand disengagement in a top-down policy context: Lessons learned from Cartagena, Colombia. Journal of Place Management and Development, 13(3), 347–363. https://doi.org/10.1108/JPMD-06-2019-0040
Blain, C., Levy, S. E., & Ritchie, J. R. B. (2005). Destination Branding: Insights and Practices from Destination Management Organizations. Journal of Travel Research, 43(4), 328–338. https://doi.org/10.1177/0047287505274646
Future Strategy Eggental 2030. (2022). Eggental 2030: The future strategy of the Dolomites holiday region. TOGETHER WE ARE... Future Strategy Eggental 2030 A Participative Strategy. https://eggental.com/en/eggental/strategy-eggental-2030
Hankinson, G. (2005). Destination brand images: A business tourism perspective. Journal of Services Marketing, 19(1), 24–32. https://doi.org/10.1108/08876040510579361
Hanna, S., Rowley, J., & Keegan, B. (2021). Place and Destination Branding: A Review and Conceptual Mapping of the Domain. European Management Review, 18(2), 105–117. https://doi.org/10.1111/emre.12433
Kotsi, F., Balakrishnan, M. S., Michael, I., & Ramsøy, T. Z. (2018). Place branding: Aligning multiple stakeholder perception of visual and auditory communication elements. Journal of Destination Marketing & Management, 7, 112–130. https://doi.org/10.1016/j.jdmm.2016.08.006
Pongsakornrungsilp, P., Pongsakornrungsilp, S., Jansom, A., & Chinchanachokchai, S. (2022). Rethinking Sustainable Tourism Management: Learning from the COVID-19 Pandemic to Co-Create Future of Krabi Tourism, Thailand. Sustainability, 14(18), Article 18. https://doi.org/10.3390/su141811375
Pongsakornrungsilp, S., Pongsakornrungsilp, P., Pusaksrikit, T., Wichasin, P., & Kumar, V. (2021). Co-Creating a Sustainable Regional Brand from Multiple Sub-Brands: The Andaman Tourism Cluster of Thailand. Sustainability, 13(16), Article 16. https://doi.org/10.3390/su13169409
Pongsakornrungsilp, S., & Schroeder, J. E. (2017). :How consumers co-create. In M. Keller, B. Halkier, & T. A. Wilska (Eds.), Routledge Handbook on Consumption. Routledge.
Stankov, U., & Gretzel, U. (2021). Digital well-being in the tourism domain: Mapping new roles and responsibilities. Information Technology & Tourism, 23(1), 5–17. https://doi.org/10.1007/s40558-021-00197-3
Xara-Brasil, D., Miadaira Hamza, K., & Marquina, P. (2018). The meaning of a brand? An archetypal approach. Revista de Gestão, 25(2), 142–159. https://doi.org/10.1108/REGE-02-2018-0029
Zenker, S., Braun, E., & Petersen, S. (2017). Branding the destination versus the place: The effects of brand complexity and identification for residents and visitors. Tourism Management, 58, 15–27. https://doi.org/10.1016/j.tourman.2016.10.008